# Quantum Computing Approaches for Vector Quantization—Current Perspectives and Developments

**DOI:** 10.3390/e25030540

**Published:** 2023-03-21

**Authors:** Alexander Engelsberger, Thomas Villmann

**Affiliations:** Saxon Institute for Computational Intelligence and Machine Learning (SICIM), University of Applied Sciences Mittweida, Technikumplatz 17, 09648 Mittweida, Germany; villmann@hs-mittweida.de

**Keywords:** vector quantization, quantum machine learning, prototype-based learning

## Abstract

In the field of machine learning, vector quantization is a category of low-complexity approaches that are nonetheless powerful for data representation and clustering or classification tasks. Vector quantization is based on the idea of representing a data or a class distribution using a small set of prototypes, and hence, it belongs to interpretable models in machine learning. Further, the low complexity of vector quantizers makes them interesting for the application of quantum concepts for their implementation. This is especially true for current and upcoming generations of quantum devices, which only allow the execution of simple and restricted algorithms. Motivated by different adaptation and optimization paradigms for vector quantizers, we provide an overview of respective existing quantum algorithms and routines to realize vector quantization concepts, maybe only partially, on quantum devices. Thus, the reader can infer the current state-of-the-art when considering quantum computing approaches for vector quantization.

## 1. Introduction

Quantum computing is an emerging research field, and the current wave of novelties is driven by advances in building quantum devices. In parallel to this hardware development, new quantum algorithms and extensions of already known methods like Grover search emerged during the last few years, for example, for graph problems [1] or image processing [2]. One field of growing interest is Quantum Machine Learning. On the one hand, we can consider quantum algorithms to accelerate classical machine learning algorithms [3,4]. On the other, machine learning approaches can be used to optimize quantum routines [5].

In this paper, we focus on the first aspect. In particular, we consider the realization of unsupervised and supervised vector quantization approaches by means of quantum routines. This focus is taken because vector quantization is one of the most prominent tasks in machine learning for clustering and classification learning. For example, (fuzzy-) *k*-means or its more modern variants *k*-means and neural gas constitute a quasi-standard in an unsupervised grouping of data, which frequently is the starting point for sophisticated data analysis to reduce the complexity of those investigations [6,7,8]. The biologically inspired self-organizing map is one of the most prominent tools for visualization of high-dimensional data, based on the concept of topology preserving data mapping [9,10,11,12]. In the supervised setting, (generalized) learning vector quantization for classification learning is a powerful tool based on intuitive learning rules, which, however, are mathematically well-defined such that the resulting model constitutes an adversarial-robust large margin classifier [13,14,15]. Combined with the relevance learning principle, this approach provides a precise analysis of the data features weighting for optimal performance, improving classification decision interpretability and, hence, allows causal inferences to interpret the feature influence for the classification decision [12,16,17].

Further, the popularity of vector quantization methods arises from their intuitive problem understanding and the resulting interpretable model behavior [8,10,18,19], which frequently is demanded for acceptance of machine learning methods in technical or biomedical applications [20,21,22]. Although these methods are of only lightweight complexity compared to deep networks, frequently sufficient performance is achieved.

At the same time, the current capabilities of quantum computers only allow a limited complexity of algorithms. Hence, the implementation of deep networks is currently not realistic apart from any mathematical challenges for realization. Therefore, vector quantization methods became attractive for the investigation of corresponding quantum computing approaches, i.e., respective models are potential candidates to run on the limited resources of a quantum device.

To do so, one can either adopt the mathematics of quantum computing for quantum-inspired learning rules to vector quantization [23], or one gets motivation from existing quantum devices to obtain quantum-hybrid approaches [24,25].

In this work, we are considering vector quantization approaches for clustering and classification in terms of their adaptation paradigms and how they could be realized using quantum devices. In particular, we discuss model adaptation using prototype shifts or median variants for prototype-based vector quantization. Further, unsupervised and supervised vector quantization is studied as a special case of set-cover problems. Finally, we also explain an approach based on Hopfield-like associative memories. Each of these adaptation paradigms comes with advantages and disadvantages depending on the task. For example, median or relational variants come into play if only proximity relations between data are available but with reduced flexibility for the prototypes [26,27]. Vector shift adaptation relates to Minkowski-like data spaces with corresponding metrics, which usually provide an obvious interpretation of feature relevance if combined with a task depending on adaptive feature weighting. Attractor networks like the Hopfield model can be used to learn categories without being explicitly trained on them [28]. The same is true of cognitive memory models [29], which have great potential for general learning tasks [30].

Accordingly, we subsequently examine which quantum routines are currently available to realize these adaptation schemes for vector quantization adaptation completely or partially. We discuss the respective methods and routines in light of the existing hardware as well as the underlying mathematical concepts. Thus, the aim of the paper is to give an overview of quantum realizations of the adaptation paradigms of vector quantization.

## 2. Vector Quantization

Vector Quantization (VQ) is a general motif in machine learning and data compression. Given a data set $X\subset R^{n}$ with |X|=N data points $x{}_{i}$, the idea of VQ is representing X using a much smaller set $W\subset R^{n}$ of vectors $w{}_{i}$, where |W|=M≪N. We will call these vectors prototypes; sometimes, they are also referred to as codebook vectors. Depending on the task, the prototypes are used for pure data representation or clustering in unsupervised learning, whereas in the supervised setting, one has to deal with classification or regression learning. A common strategy is the nearest prototype principle for a given data x realized using a winner takes all rule (WTA-rule), i.e.,
(1)$$s\left(x\right)={\mathrm{argmin}}_{j=1,\ldots ,M}\left(d\left(x,w{}_{j}\right)\right)\in \left\{1,\ldots ,M\right\}$$
for a given dissimilarity measure *d* in $R^{n}$ and where $w{}_{s}$ is denoted as the winning prototype of the competition. Hence, an appropriate choice of the metric *d* in use seriously influences the outcome of the VQ approach. Accordingly, the receptive fields of the prototypes are defined as
$$R\left(w{}_{j}\right)=\left\{x{}_{i}\in X|s\left(x{}_{i}\right)=j\right\}$$
with $X=\cup_{j=1}^{M}R\left(w{}_{j}\right)$.

### 2.1. Unsupervised Vector Quantization

Different approaches are known for optimization of the prototype set W for a given dataset X, which are briefly described in the following. In the unsupervised setting, no further information is given.

#### 2.1.1. Updates Using Vector Shifts

We suppose an energy function
$$E_{VQ}\left(X,W\right)=\sum_{i=1}^{N}E_{VQ}\left(x{}_{i},W\right)$$
with local errors $E_{VQ}\left(x{}_{i},W\right)$ to be assumed as differentiable with respect to the prototypes and, hence, the dissimilarity measure *d* is also supposed to be differentiable. Further, the prototype set W is randomly initialized. Applying the stochastic gradient descent learning for prototypes, we obtain the prototype update
$$\Delta w{}_{j}\propto -\frac{\partial E_{VQ}\left(x{}_{i},W\right)}{\partial d\left(x{}_{i},w{}_{j}\right)}\cdot \frac{\partial d\left(x{}_{i},w{}_{j}\right)}{\partial w{}_{j}}$$
for a randomly selected sample $x{}_{i}\in X$ [31]. If the squared Euclidean distance $d_{E}\left(x,w{}_{j}\right)={\left(x-w{}_{j}\right)}^{2}$ is used as the dissimilarity measure, the update obeys a vector shift
$$\frac{\partial d_{E}\left(x{}_{i},w{}_{j}\right)}{\partial w{}_{j}}=-2\left(x-w{}_{j}\right)$$
attracting the prototype $w{}_{j}$ towards the presented data $x{}_{i}$.

Prominent in those algorithms is the well-known online *k*-means or its improved variant, the neural gas algorithm, which makes use of prototype neighborhood cooperativeness during training to accelerate the learning process as well as for initialization insensitive training [8,32].

Further, note that similar approaches are known for topologically more sophisticated structures like subspaces [33].

#### 2.1.2. Median Adaptation

In median VQ approaches, the prototypes are restricted to be data points, i.e., for a given $w{}_{j}$ exists a data sample $x{}_{i}$ such that $w{}_{j}=x{}_{i}$ is valid. Consequently, W⊂X holds. The inclusion of a data point into the prototype set can be represented using a binary index variable; using this representation, a connection to the binary optimization problem becomes apparent.

Optimization of the prototype set W can be achieved with a restricted expectation maximization scheme (EM) of alternating optimization steps. During the expectation step, the data are assigned to the current prototypes, whereas in the maximization step, the prototypes are re-adjusted with the median determination of the current assignments. The corresponding counterparts of neural gas and *k*-means are median neural gas and *k*-medoids, respectively [26,34].

#### 2.1.3. Unsupervised Vector Quantization as a Set-Cover Problem Using ϵ-Balls

Motivated by the notion of receptive fields for VQ, an approach based on set covering was introduced. In this scenario, we search for a set $W_{\epsilon }\subset R^{n}$ to represent the data X through prototype-dependent ϵ-balls
(2)$$B_{\epsilon }\left(w{}_{j}\right)=\left\{x\in R^{n}|d\left(x,w{}_{j}\right)<\epsilon \right\}$$
for prototypes $w{}_{j}\in W_{\epsilon }$. More precisely, we consider the ϵ-restricted receptive fields of prototypes
$$R_{\epsilon }\left(w{}_{j}\right)=\left\{x{}_{i}\in X|s_{\epsilon }\left(x{}_{i}\right)=j\right\}$$
for a given configuration $W_{\epsilon }$, where
$$s_{\epsilon }\left(x\right)=\left\{\begin{array}{cc} j & \mathrm{if}\;s\left(x\right)=j\;\mathrm{and}\;d\left(x,w{}_{j}\right)<\epsilon \\ \emptyset & \mathrm{else} \end{array}\right.$$
is the ϵ-restricted winner determination, and ‘∅’ denotes the no-assignment-statement. Hence, $R_{\epsilon }\left(w{}_{j}\right)$ consists of all data $x{}_{i}\in X$ covered by an ϵ-ball such that we have $R_{\epsilon }\left(w{}_{j}\right)\subseteq B_{\epsilon }\left(w{}_{j}\right)$.

The task is to find a minimal prototype set $W_{\epsilon }$ such that the respective cardinality $M_{\epsilon }$ is minimum while the unification $B_{\epsilon }\left(W_{\epsilon }\right)=\cup_{j=1}^{M_{\epsilon }}B_{\epsilon }\left(w{}_{j}\in W_{\epsilon }\right)$ is covering the data X, i.e., $X\subseteq B_{\epsilon }\left(W_{\epsilon }\right)$ has to be valid. A respective VQ approach based on vector shifts is proposed [35].

The set-covering problem becomes much more difficult if we restrict the prototypes $w{}_{j}\in W_{\epsilon }$ to be data samples $x{}_{i}\in X$, i.e., $W_{\epsilon }\subset X$. This problem is known to be NP-complete [36]. A respective greedy algorithm was proposed [37]. It is based on a kernel approach, taking the kernel
$$\kappa_{\epsilon }\left(x{}_{j},x{}_{i}\right)=\left\{\begin{array}{cc} 1 & \mathrm{if}\;d_{E}\left(x{}_{j},x{}_{i}\right)<\epsilon \\ 0 & \mathrm{else} \end{array}\right.$$
as an indicator function. The kernel $\kappa_{\epsilon }$ corresponds to a mapping
$$\phi_{\epsilon }\left(x{}_{i}\right)={\left(\kappa_{\epsilon }\left(x_{1},x{}_{i}\right),\ldots ,\kappa_{\epsilon }\left(x{}_{N},x{}_{i}\right)\right)}^{T}\in R^{N}$$
known as kernel feature mapping [38]. Introducing a weight vector $w\in R^{N}$, the objective
$$E_{q,\epsilon }\left(X\right)=\underset{\mathrm{subject}\mathrm{to}\;{\langle w,\phi_{\epsilon }\left(x{}_{i}\right)\rangle }_{E}\geq 1\;\forall i}{\min_{w\in R^{N}}{∥w∥}_{q}}$$
appears as the solution of a minimum problem depending on the parameter *q* in the Minkowski-norm ${∥w∥}_{q}$. For the choice q=0, we would obtain the original problem. However, for q=1, good approximations are achieved and can be done efficiently using linear programming [37]. After optimization, the data samples $x{}_{i}$ with $w_{i}\approx 1$ serve as prototypes. The respective approach can be optimized online based on neural computing [39,40].

#### 2.1.4. Vector Quantization by Means of Associative Memory Networks

Associative memory networks have been studied for a long time [9,41]. Among them, Hopfield networks (HNs) [41,42] have gained a lot of attraction [30,43,44]. In particular, the strong connection to physics is appreciated [45]; it is related to other optimization problems as given in Section 3.2.3.

Basically, for $X\subset R^{n}$ with cardinality *N*, HNs are recurrent networks of *n* bipolar neurons $s_{i}\in \left\{-1,1\right\}$ connected to each other by the weights $W_{ij}\in R$. All neurons are collected in the neuron vector $s={\left(s_{1},\ldots ,s_{n}\right)}^{T}\in {\left\{-1,1\right\}}^{n}$. The weights are collected in the matrix $W\in R^{m\times m}$ such that to each neuron $s_{i}$ belongs a weight vector $w_{i}$. The matrix W is assumed to be symmetric and hollow, i.e., $W_{ii}=0$. The dynamic of the network is
(3)$$s_{i}=\mathrm{sgn}\left({\langle s,w_{i}\rangle }_{E}-\theta_{i}\right)$$
where
$$\mathrm{sgn}\left(z\right)=\left\{\begin{array}{cc} 1 & \mathrm{if}\;z\geq 0\\ -1 & \mathrm{else} \end{array}\right.$$
is the standard signum function of z∈R and $\theta_{i}$ is the neuron-related bias generating the vector $\theta ={\left(\theta_{1},\ldots ,\theta_{n}\right)}^{T}$. According to the dynamic (Equation 3), the neurons in an HN are assumed to be perceptrons with the signum function as activation [46,47]. Frequently, the vectorized notation
(4)$$s^{\prime }=\mathrm{sgn}\left[Ws\right]-\theta$$
of the dynamic (Equation 3) is more convenient, emphasizing the asynchronous dynamic. The network minimizes the energy function
(5)$$E_{H}\left(s\right)=-\frac{1}{2}s^{T}Ws+{\langle s,\theta \rangle }_{E}$$
in a finite number of steps, with an asynchronous update dynamic [45].

For given bipolar data vectors $x{}_{i}\in X$ with dataset cardinality N≪n, the matrix $W\in R^{n\times n}$ is obtained with the entries
(6)$$W_{ij}=\frac{1}{N}\sum_{k=1}^{N}{\left[x{}_{k}\right]}_{i}\cdot {\left[x{}_{k}\right]}_{j}=\frac{1}{N}\sum_{k=1}^{N}x{}_{k}\cdot x_{k}^{T}-I$$
where $I\in R^{n\times n}$ is the identity matrix. This setting can be interpreted as Hebbian learning [45]. Minimum solutions $s^{*}\in {\left\{-1,1\right\}}^{n}$ of the dynamic (Equation 7) are the data samples $x{}_{i}$. Thus, starting with arbitrary vectors s, the network always relaxes to a stored pattern $x{}_{i}$ realizing an association scheme if we interpret the starting point as a noisy pattern. The maximum storage capacity of an HN is limited to $c_{s}=\frac{N}{n}$ patterns with $c_{s}\leq c_{\max}∼0.138$. Dense Hopfield networks (DHNs) are generalizations of HNs with general data patterns $x{}_{i}\in X\subset R^{n}$ having a much greater storage capacity of $c_{\max}=1$ [48].

For the unsupervised VQ, an HN can be utilized using a kernel approach [49]: Let
$$p\left(x\right)=\frac{1}{N}\sum_{i=1}^{N}\kappa_{\phi }\left(x,x{}_{i}\right)$$
be an estimate of the underlying data density $R^{n}$ based on the samples $X\subset R^{n}$ with |X|=N. Analogously,
$$\hat{q}\left(x\right)=\frac{1}{M}\sum_{j=1}^{M}\kappa_{\phi }\left(x,w{}_{j}\right)\approx \frac{1}{N}\sum_{i=1}^{N}\kappa_{\phi }\left(x,x{}_{i}\right)\cdot a_{i}$$
is an estimate of the data density $R^{n}$ based on the *M* prototypes $W\subset R^{n}$. The density $\hat{q}\left(x\right)$ can be approximated with
$$q\left(x\right)=\frac{1}{N}\sum_{i=1}^{N}\kappa_{\phi }\left(x,x{}_{i}\right)\cdot a_{i}$$
for assignment variables $a_{i}\in \left\{0,1\right\}$ collected in the vector $a={\left(a_{1},\ldots ,a_{N}\right)}^{T}$ with the constraint $\sum_{i=1}^{N}a_{i}=M$. According to the theory of kernels, the kernel $\kappa_{\phi }$ relates to a map $\phi :R^{n}\to H$, where H is a reproducing kernel Hilbert space (RKHS) endowed with an inner product ${\langle \cdot |\cdot \rangle }_{H}$ such that
$${\langle \phi \left(x\right)|\phi \left(x^{\prime }\right)\rangle }_{H}=\kappa_{\phi }\left(x,x^{\prime }\right)$$
holds [38].

For a good representation of X with the prototype W, it is possible to minimize the quantity
$$\hat{D}\left(X,W\right)={∥E_{X}\left[\phi \right]-E_{W}\left[\phi \right]∥}_{H},$$
where $E_{X}\left[\phi \right]$ and $E_{W}\left[\phi \right]$ are the expectations of ϕ based on the sets X and W, respectively, using the densities $p\left(x\right)$ and $q\left(x\right)$ [49]. We obtain
$$\hat{D}\left(X,W\right)=\frac{1}{N^{2}}1^{T}\Phi 1+\frac{1}{M^{2}}a^{T}\Phi a-\frac{2}{N\cdot M}1^{T}\Phi a$$
with $1={\left(1,\ldots ,1\right)}^{T}\in R^{N}$, $\Phi \in R^{N\times N}$ and $\Phi_{ij}=\kappa_{\phi }\left(x{}_{i},x{}_{j}\right)$. Because the first term $1^{T}\Phi 1$ does not depend on the assignment, minimization of $D\left(X,W\right)$ with respect to the assignment vector a is equivalent to a minimization of
$$D\left(X,W\right)=\frac{1}{M^{2}}a^{T}\Phi a-\frac{2}{N\cdot M}1^{T}\Phi a$$
subject to the constraint ${\langle 1^{T},a\rangle }_{E}=M$ or, equivalently, ${\left(1^{T}\cdot a-M\right)}^{2}=0$ such that it constitutes a Lagrangian optimization with the multiplier $\lambda_{L}$. Transforming the binary vector a using s=2·a−1 into a bipolar vector, the constraint minimization problem is reformulated as
(7)$$s^{*}={\mathrm{argmin}}_{s\in {\left\{-1,1\right\}}^{N}}\left(s^{T}Qs+{\langle s,q\rangle }_{E}\right)$$
with
$$Q=\frac{1}{4}\left(\frac{1}{M^{2}}\Phi -\lambda_{L}1\cdot 1^{T}\right)$$
and
$$q=\frac{1}{2}\left(\frac{1}{M^{2}}\Phi -\lambda_{L}1\cdot 1^{T}\right)\cdot 1-\left(\frac{2}{M\cdot N}\Phi^{T}\cdot 1+2\cdot \lambda_{L}\cdot M\cdot 1\right),$$
both depending on the Lagrangian multiplier $\lambda_{L}$. Thus, the problem (Equation 7) can be translated into the HN energy $E\left(s\right)$ with m=M, θ=q,
$$W=-2\cdot Q-\frac{\lambda_{L}}{2\cdot M^{2}}\cdot I,$$
where $I\in R^{N\times N}$ is the unity matrix and $s^{*}$ obtained using the HN dynamic (Equation 5).

Complex-valued Hopfield networks (CHN) are extending the HN concept to complex numbers [50]. For this purpose, the symmetry assumption for the weights $W_{ij}$ is transferred to the Hermitian symmetry $W_{ij}={\bar{W}}_{ij}$ of the conjugates. As in the real case, the complex dynamic is structurally given as in (Equation 3) but replacing the real inner product using the complex-valued Euclidean inner product and, as the consequence of that, replacing the signum function $\mathrm{sgn}\left(z\right)$, too. Instead of this, the modified ‘signum’ function
$$\mathrm{csgn}\left(z\right)=\left\{\begin{array}{cc} e^{0\cdot i}=1 & \mathrm{if}\;0\leq \arg\left(z\right)<\varpi_{R}\\ e^{1\cdot i\cdot \varpi_{R}} & \mathrm{if}\;\varpi_{R}\leq \arg\left(z\right)<2\varpi_{R}\\ \vdots & \vdots \\ e^{\left(R-1\right)\cdot i\varpi_{R}} & \left(R-1\right)\cdot \varpi_{R}\leq \arg\left(z\right)\leq R\cdot \varpi_{R} \end{array}\right.$$
for complex-valued *z* is used, with *R* being the resolution factor for the phase range delimitation [51]. Thus, $\arg\left(z\right)$ is the phase angle of *z* and $\varpi_{R}=\frac{2\pi }{R}$ determines the partition of the phase space. The Hebbian learning rule (Equation 6) changes to
$$W_{ij}=\frac{1}{N}\sum_{k=1}^{N}{\left[x{}_{k}\right]}_{i}\cdot {\bar{\left[x{}_{k}\right]}}_{j}$$
and the energy of the CHN is obtained as
$$E_{H}\left(s\right)=-\frac{1}{2}s^{T}Ws$$
for zero bias, which delivers
$$s^{\prime }=\mathrm{csgn}\left[Ws\right]$$
as the corresponding dynamic in complete analogy to (Equation 4). Note, for the resolution R=2, the usual HN is obtained.

### 2.2. Supervised Vector Quantization for Classification Learning

For classification learning VQ, we assume that the training data $x{}_{i}\in X\subset R^{n}$ are endowed with a class label $y{}_{i}=c\left(x{}_{i}\right)\in C=\left\{1,\ldots ,C\right\}$. Besides the widespread deep networks, which are powerful methods in classification learning but do not belong to VQ algorithms, support vector machines (SVMs) are promising robust classifiers optimizing the separation margin [52]. However, the support vectors, which determine the class borders of the problem, sometimes are interpreted as prototypes such that SVM could be taken as a supervised prototype classifier, too [53]. However, we do not focus on SVM here.

#### 2.2.1. Updates Using Vector Shifts

Prototype-based classification learning based on vector shifts is dominated by the family of learning vector quantizers (LVQ), which was heuristically motivated and already introduced in 1988 [54]. These models assume that for each prototype $w{}_{j}\in W$, we have an additional class label $c\left(w{}_{j}\right)\in C$, such that at least one prototype is dedicated to each class. For a given training data pair $\left(x{}_{i},y{}_{i}\right)$, let $w^{+}$ denote the best matching prototype $w{}_{s}$ determined with the WTA-rule (Equation 1) with additional constraint that $y{}_{i}=c\left(w{}_{s}\right)$ and $d^{+}\left(x{}_{i}\right)=d\left(x{}_{i},w^{+}\right)$ denotes the respective dissimilarity. Analogously, $w^{-}$ is the best matching prototype $w_{s^{\prime }}$ with the additional constraint that $y{}_{i}\neq c\left(w_{s^{\prime }}\right)$ and $d^{-}\left(x{}_{i}\right)=d\left(x{}_{i},w^{-}\right)$. The basic principle in all LVQ models is that if $d=d_{E}$ is the squared Euclidean distance, the prototype $w^{+}$ is attracted by the presented training data sample $x{}_{i}$ whereas $w^{-}$ is repelled. Particularly, we have
$$\Delta w^{+}\propto -2\cdot \left(x{}_{i}-w^{+}\right)\;\mathrm{and}\;\Delta w^{-}\propto -2\cdot \left(w^{-}-x{}_{i}\right),$$
which is known as the *attraction-repulsing-scheme* (ARS) of LVQ.

The heuristic LVQ approach can be replaced by an approach grounded on a cost function [55], which is based on the minimization of the approximated classification error
(8)$$E_{GLVQ}\left(X,W\right)=\sum_{i=1}^{N}E_{GLVQ}\left(x{}_{i},W\right)$$
with local errors
$$E_{GLVQ}\left(x{}_{i},W\right)=f_{\theta }\left(\mu \left(x{}_{i}\right)\right)$$
evaluating the possible classification mismatch for a given data sample $x{}_{i}$. Thereby,
$$\mu \left(x{}_{i}\right)=\frac{d^{+}\left(x{}_{i}\right)-d^{-}\left(x{}_{i}\right)}{d^{+}\left(x{}_{i}\right)+d^{-}\left(x{}_{i}\right)}\in \left[-1,+1\right]$$
is the so-called classifier function resulting in non-positive values when the sample $x{}_{i}$ would be incorrectly classified. The function
$$f_{\theta }\left(z\right)=\frac{1}{1+\exp\left(-z\cdot \theta \right)}$$
is the sigmoid, approximating the Heaviside function
$$H\left(z\right)=\left\{\begin{array}{cc} 1 & \mathrm{if}\;z>0\\ 0 & \mathrm{else} \end{array}\right.$$
but keeping the differentiability. Following this definition, the updates for $w^{+}$ and $w^{-}$ in (Equation 8) are obtained as
$$\Delta w^{\pm }\propto -2\cdot \frac{f_{\theta }^{\prime }\left(\mu \left(x{}_{i}\right)\right)\cdot d^{\mp }\left(x{}_{i}\right)}{{\left(d^{+}\left(x{}_{i}\right)+d^{-}\left(x{}_{i}\right)\right)}^{2}}\cdot \left(x{}_{i}-w^{\pm }\right),$$
realizing an ARS [55].

This variant of LVQ is known as Generalized LVQ and is proven to be robust against adversarials [14]. For variants including metric learning, we refer to [12]. Complex-valued GLVQ using the Wirtinger calculus for gradient calculations are considered [56].

Learning on topological structures like manifolds and subspaces follows the same framework, considering attraction and repulsing more general in the respective vector spaces [57,58]. An interesting variant, where the prototypes are spherically adapted according to an ARS to keep them on a hypersphere, was proposed—denoted as Angle-LVQ [59].

#### 2.2.2. Median Adaptation

Median LVQ-like adaptation of prototypes for classification learning is possible [27]. This variant is based on an alternating optimization scheme similar to that of medoid *k*-means and median neural gas but adapted to the classification-restricted setting.

#### 2.2.3. Supervised Vector Quantization as a Set-Cover Problem Using ϵ-Balls

Another classification scheme can be based on prototype selection out of the training samples and ϵ-balls [60]. In analogy to ϵ-balls for prototypes defined in (Equation 2), Data-dependent counterparts are defined as
$$B_{\epsilon }\left(x{}_{i}\right)=\left\{x{}_{j}|d\left(x{}_{i},x{}_{j}\right)<\epsilon \right\}$$
the union of which trivially covers X. The classification problem is then decomposed into separate cover problems per class, as discussed in Section 2.1.3. For this purpose, each ϵ-ball gets a local cost based on the number of covered points, punishing false classified points using a penalty
$$\frac{1}{\left|X\right|}+\left|B_{\epsilon }\left(x{}_{i}\right)\cap \left(X\setminus X_{c}\right)\right|$$
where $X_{c}$ is the set of all data points with the same class as $x{}_{i}$. Combined with a unit cost for not covering a point, a prize-collecting set-cover problem is defined that can be transformed into a general set-cover problem. Hence, as an objective, the number of covered and correctly classified data points has to be maximized while keeping the overall number of prototypes low. We refer to [60,61] for detailed mathematical analysis. In particular, a respective approach is presented [61], being similar to the optimization scheme from support vector machines [52].

#### 2.2.4. Supervised Vector Quantization by Means of Associative Memory Networks

Classification by means of associative memory networks is considered classification using Hopfield-like networks [30]. An approach based on spiking neurons instead of perceptron-like neurons in HNs as depicted in (Equation 3) was presented using a classical spike-timing-dependent-plasticity (STDP) rule for learning to adapt HNs for classification learning [62].

In contrast, a modified HN for classification can be used [63]. We suppose a dataset $X\subset R^{n}$ consisting of *N* samples distributed to *C* classes. A template vector $\xi^{c}\in R^{N}$ is introduced for each class c∈C with $\xi_{i}^{c}=1$ if $c=y{}_{i}$ and $\xi_{i}^{c}=-1$, otherwise. The states of neurons $s_{k}$ are extended to be $s_{k}\in \left\{-1,1,0\right\}$ for k=1,…,N constituting the vector s. We consider a diluted version of the Hopfield model, where the weight matrix $W\in R^{N\times N}$ is considered to be
$$W_{ij}=\left\{\begin{array}{cc} -\frac{C}{N} & \mathrm{if}\;y{}_{i}=y{}_{j}\\ \frac{C}{2\cdot N}\sum_{c=1}^{C}\xi_{i}^{c}\cdot \xi_{j}^{c}+2-C & \mathrm{else} \end{array}\right.$$
realizing a slightly modified Hebb-rule compared to (Equation 6). The dynamic is still (Equation 3) as in the usual Hopfield model. However, if a switch from $s_{k}=1$ to $s_{k}=-1$ is observed as the result of the dynamic, $s_{k}=0$ is set to switch of the respective neuron [63].

## 3. Quantum Computing—General Remarks

In the following, we use the terms quantum and classical computer to describe whether a machine exploits the rules of quantum mechanics to do its calculations or not.

### 3.1. Levels of Quantum Computing

Quantum Algorithms can be classified into at least three levels: quantum-inspired, quantum-hybrid, and quantum(-native), with increasing dependence on the capabilities of quantum computers.

Working with the mathematical foundation of quantum computing may reveal new insides into classical computing. In this view, classical algorithms appear in a new form, which is not dependent on the execution on real quantum computers but incorporates the mathematical framework of quantum systems to obtain specific variants of the original algorithm. This category of algorithms is called *quantum-inspired algorithms*. For example, in supervised VQ, an approach inspired by quantum mechanics has been developed, based on standard GLVQ, but now adapted to problems where both the data and the prototypes are restricted to the unit sphere [23]. Thus, this algorithm shows similarities to the already mentioned classical Angle LVQ. However, in contrast to this, here, the sphere is interpreted as a Bloch sphere, and the prototype adaptation follows unitary transformations.

While quantum-inspired algorithms only lend the mathematical background of quantum computing, *quantum-hybrid algorithms* use a quantum device as a coprocessor to accelerate the computations. The quantum chip is also referred to as Quantum Processing Unit (QPU) [64]. The QPU is used to solve expensive computational tasks like searching or high-dimensional distance calculations, whereas all other program logic, like data loading or branching, is done using a classical machine.

The quantum-hybrid algorithm can also be defined in more rigorous terms. That is, a quantum-hybrid algorithm requires, for example, “non-trivial amounts of both quantum and classical computational resources” [64]. Following this definition, classical control elements, like repetition until a valid state is found, are not considered hybrid systems.

Finally, as *quantum-native algorithms*, we would like to denote those algorithms that run entirely on a quantum machine after the data is loaded into it. Because of the limitations of the present hardware generation, their physical implementation is not feasible so far, and therefore, ongoing research is often focused on quantum-hybrid strategies under the prevailing circumstances.

### 3.2. Paradigms of Quantum Computing

Quantum Physics can be harnessed for computing using different kinds of computing paradigms. Currently, there are two major paradigms intensively investigated and discussed for applications: *Gate-based* and *adiabatic* quantum computing. It can be shown that both paradigms are computationally equivalent [65]. Nevertheless, it is interesting to consider these two approaches separately, as they lead to different problems and solutions that are better suited for their underlying hardware. There are several other paradigms, such as measurement-based and topological quantum computing. We will not focus on them in this paper but concentrate on gate-based and adiabatic methods as the most important.

#### 3.2.1. Gate Based Quantum Computing and Data Encoding

Classical computers store information as bits that are either 0 or 1. The smallest unit of a quantum computer is called a *qubit* [66]. It can represent the classical states as |0〉 and |1〉. Besides these basis states, every linear combination of the form
$$|\psi \rangle =a|0\rangle +b|1\rangle \;\;\mathrm{with}\;\;a,b\in {C:\;|a|}^{2}+{\left|b\right|}^{2}=1.$$
is a valid state of a qubit. If ab≠0, the qubit is in a so-called superposition state. Alternatively, the qubit can also be written as a wave function
$$\psi =\left(\begin{array}{c} a\\ b \end{array}\right)$$
with the normalization constraint for *a* and *b* remains to be valid.

When measured, the qubit turns into one of the two classical states according to the probabilities ${\left|a\right|}^{2}$ and ${\left|b\right|}^{2}$, respectively. In other words, during measurement, the state changes into the observed one; this effect is called the collapse of the wave function. To get the probabilistic information about *a* and *b*, it is, in general, necessary to measure a state multiple times. Because of the collapsing wave function and the so-called no-cloning theorem, this can only be achieved by preparing a qubit multiple times in the same known manner [67].

A collection of qubits is called a quantum register. To represent the state of a quantum register, we write |i〉 if the quantum register is the binary representation of the non-negative integer *i*. The wave function for a register containing *N* qubits is represented by a normalized complex vector of length $2^{N}$:$$\psi =\sum_{i=0}^{2^{N}-1}\psi_{i}|i\rangle =:|\psi \rangle \;\;\mathrm{with}\;\;\sum_{i=0}^{2^{N}-1}{\left|\psi_{i}\right|}^{2}=1$$
with the complex amplitudes $\psi_{i}\in C$. For independent qubits, the state of the register is the tensor product of its qubits, and otherwise, we say that the qubits are entangled. For a deeper introduction to the mathematics of qubits and quantum processes, we recommend [66,68] to the reader.

##### Basis Encoding

In classical computing, information is represented by a string of bits. Obviously, it is possible to use coding schemes such as floating-point numbers to represent more complex data structures, too. These methods can also be used on a quantum computer without the application of superposition or entanglement effects. However, taking these quantum effects into account enables quantum-specific coding methods.

Besides storing a single bit-sequence, a superposition of multiple sequences of the same length can be stored in a single quantum register as
$$\sum_{i=0}^{2^{N}-1}w_{i}|x{}_{i}\rangle ,$$
where $w_{i}$ is the weight of the sequence $x{}_{i}$. Thus, the measurement probability $p_{i}={\left|w_{i}\right|}^{2}$ is valid. Algorithms that run on basis encoding often amplify valid solution sequences of a problem by using interference patterns of the complex phases of various $w_{i}$.

A state in this basis encoding scheme can be initialized using the Quantum Associative Memory Algorithm [69].

##### Amplitude Encoding

In the amplitude encoding scheme, for a given complex vector x, its entries are encoded inside the amplitudes $\psi_{i}$ of a quantum register. For this purpose, first, the vector has to be normalized, choosing a normalization that limits the impact on a given task with data distortion. If the vector size is not a power of two, zero padding is applied. We can now, in the second step, initialize a quantum state with $\psi_{i}={\hat{x}}_{i}$ for the normalized and padded vector $\hat{x}$. A state in this amplitude encoding can be generated using a universal initialization strategy [70].

A highly anticipated, but still not realized, hardware concept is the QRAM [71]. It is key for the speedup of many quantum algorithms, but its viability remains open. Still, its future existence is commonly assumed.

##### Gate-Based Quantum Paradigm

A common concept for quantum computing is the gate notation, originally introduced by Feynman [72]. In this notation, the time evolution of a qubit is represented by a horizontal line. Evolution is realized by quantum gates that are defined by a unitary matrix applied to a number of qubits. Unitary matrices are vector norm preserving and, therefore, they also preserve the property of being a wave function [68]. Combined with measurement parts, we get a quantum circuit description. A quantum circuit can be seen as the quantum counterpart to a logical circuit.

We will utilize the bundle notation given in Figure 1a to combine multiple qubits into quantum registers. In some quantum routines, the concept of branching is used, where the computation is only continued if measuring a qubit achieves a certain result. In Figure 1b, the output of the circuit is only considered if the qubit is measured as zero. Finally, we use the arrow notation in Figure 1c to represent garbage states. They do not contain usable information anymore, but are still entangled qubits related to the system. We use the term reset over garbage, or simply garbage problem, to emphasize the necessity of appropriately handling this situation. Generally, since garbage states are usually entangled, they cannot be reused, and hence, one resets them using un-computation, i.e., setting them to zero. Of course, the details of the garbage problem are dependent on the circuit in use.

#### 3.2.2. Adiabatic Quantum Computing and Problem Hamiltonians

Adiabatic Quantum Computing (AQC) is a computing thought emerging from the adiabatic theorem [73]. It is based on Hamiltonians, which describe the time evolution of the system inside the Schrödinger Equation [74]. A Hamiltonian is realized as a Hermitian matrix H. For adiabatic computing, the corresponding eigenequation is considered. Due to the Hermitian property, all eigenvalues are real, and hence, they can be ordered. They are known as energy levels, with the smallest one being called the ground state.

In this view, if a problem solution can be transformed into the ground state of a known problem Hamiltonian $H_{P}$, the adiabatic concept defines a quantum routine that finds this ground state [75]. It starts from an initial Hamiltonian $H_{B}$, with a known and simple ground state preparation. On this initial state, usually the equal superposition of all possible outcomes, a time-dependent Hamiltonian
$$H\left(t\right)=\frac{1-t}{T}H_{B}+\frac{t}{T}H_{P},$$
that slowly shifts from $H_{B}$ to $H_{P}$, is applied over a time period *T*. The adiabatic theorem ensures that if the period *T* is sufficiently large, the system tends to stay in the ground state of the gradually changing Hamiltonian. After application, the system is in the ground state of $H_{P}$ with a very high probability. For a given problem, the final ground state is the single solution or a superposition of all valid solutions. One solution is then revealed by measuring the qubits. If AQC is run on hardware, manufacturers use the term *quantum annealing* instead to underline the noisy execution environment. The capabilities of a quantum annealer are restricted to optimization problems by their design; it is not possible to use the current generation for general quantum computing that is equivalent to the gate-based paradigm.

The dynamic AQC can be approximated using discrete steps on a gate-based quantum computer [76].

#### 3.2.3. QUBO, Ising Model, and Hopfield Network

Depending on the theoretical background an author is coming from, three main kinds of optimization problems are often encountered in the literature that share similar structures and can be transformed into each other. First, the Quadratic Unconstrained Binary Optimization problem (QUBO) is the optimization of a binary vector $x\in {\left\{0,1\right\}}^{n}$ for a cost function
$$y=x^{T}Ax=\sum_{i\leq j}A_{ij}x_{i}x_{j}$$
with a real valued upper triangle matrix A. Second, the Ising model is motivated by statistical physics and based on spin variables, which can be in state −1 and 1 [67]. The objective of the Ising model is finding a spin vector $x\in {\left\{-1,1\right\}}^{n}$, which optimizes
$$y=\sum_{i}h_{i}x_{i}+\sum_{i<j}J_{ij}x_{i}x_{j}$$
with pairwise interactions $J_{ij}$ and an external field $h_{i}$. A Quantum Annealer is a physical implementation of the Ising Model with limited pairwise interactions. Binary variables *b* can be transformed into spin variables *s* and vice versa by the relation
$$b=\frac{1+s}{2},$$
making the Ising model and QUBO mathematically equivalent. Third, the Hopfield energy function (Equation 5) was introduced as an associative memory scheme based on Hebbian learning [42,45]. Its discrete form is equivalent to the Ising model if the neurons in this associative memory model are interpreted as bipolar. All models are NP-hard and can, therefore, in theory, be transformed into all NP problems. For a broad list of these transformations, we recommend [77].

### 3.3. State-of-the-Art of Practical Quantum Experiments

In the last few years, the size of commercial gate-based general-purpose quantum computers did grow from 27 (2019 IBM Falcon) to 433 qubits (2022 IBM Osprey). Thus, the hardware has grown from simple physical demonstrators to machines called Noisy Intermediate-Scale Quantum Computer (NISQ) [78]. However, this hardware generation is still severely restricted by its size and a high error rate.

The latter problem could be solved using quantum error correction or quantum error mitigation schemes. Quantum error mitigation is a maturing field of research, with frameworks like Mitiq [79] being published. Common to most of these mitigation techniques is that a higher number of physical qubits is required to obtain a single logical qubit with a lower noise level, making the size problem the major one.

Different physical realizations of quantum computer hardware exist; we can only give some examples. Realizations based on superconducting qubits for gate-based (IBM Q System One) and for adiabatic (D-Wave’s Advantage QPU) are available. Further, quantum devices that are based on photons (Xanadu’s Borealis) or trapped ions (Honeywell System Model H1) exist.

For small toy application problems, it is possible to simulate the behavior of a quantum computer by means of a classical computing machine. Particularly, single steps of the gate-based concept can be simulated using respective linear algebra packages. Otherwise, circuits can be built in quantum computing frameworks, like IBM’s Qiskit [80] or Xanadu’s Pennylane [81]. It is also possible to simulate AQC behavior for evolving quantum systems [82]. Quantum machines that are available through online access allow observing the influence of noise on quantum algorithms based on tiny examples.

## 4. Quantum Approaches for Vector Quantization

The field of quantum algorithms for VQ is currently a collection of quantum routines that can solve particular sub-tasks than complete algorithms available for practical applications. Combinations of those routines with machine learning approaches beside traditional VQ-learning have been proposed for different fields, for example, in connection to support vector machines [83] or generative adversarial networks [84].

In this section, we present two strategies to combine classical prototype-based vector quantization principles for VQ with appropriate quantum algorithms. Thereby, we roughly follow the structure for unsupervised/supervised vector quantization learning, as explained in the Section 2.1 and Section 2.2.

By doing so, we can replace, on the one hand, single routines in the (L)VQ learning schemes using quantum counterparts. On the other, if we can find a VQ formalism that is based on a combinatorial problem, preferably a QUBO, several quantum solvers have already been proposed and, hence, could be used to tackle the problem.

### 4.1. Dissimilarities

As previously mentioned at the beginning of Section 2, the choice of the dissimilarity measure in vector quantization is crucial and influences the outcome of the learning. This statement remains true also for quantum vector quantization approaches. However, in the quantum algorithm context, the dissimilarity concepts are closely related to the coding scheme as already discussed in Section 3.2. Here it should be explicitly mentioned that the coding can be interpreted as *quantum feature mapping* of the data into a Hilbert space, which is the Bloch-sphere [4,23]. Hence, the dissimilarity calculation represents distance calculations in the Bloch sphere. However, due to this quantum feature mapping, the interpretation of the vector quantization algorithm with respect to the original data space may be limited, whereas, within the Bloch sphere (Hilbert space), the prototype principle and interpretation paradigms remain true. Thereby, the mapping here is analogous to the kernel feature mapping in support vector machines [38] as pointed out frequently [85,86,87].

Two quantum routines are promising for dissimilarity calculation: the SWAP test [88] and the Hadamard test, used in quantum classification tasks [89,90]. Both routines generate a measurement that is related to the inner product of two normalized vectors in the Bloch sphere. These input vectors are encoded using amplitude encoding. The strategies differ in their requirements for state preparation.

The *SWAP test* circuit is shown in Figure 2. This circuit is sampled multiple times. From these samples, the probability distribution of the ancilla bit is approximated, which is connected to the Euclidean inner product by
$$p_{a}\left(|0\rangle \right)=\frac{1}{2}\left(1+{\left|\langle x|w{}_{k}\rangle \right|}^{2}\right).$$

Thus, we can calculate the inner product from the estimated probability and, hence, from that, the Euclidean distance.

Another but similar approach [89,90], which is based on the Hadamard gate, sometimes denoted as a (modified) *Hadamard test*, is shown in Figure 3. For this circuit, the probability of measuring the ancilla in zero state is
pa(|0〉)=121+Re{〈x|wk〉}.

Due to the superposition principle, it is possible to run these tests in parallel on different inputs. This method was demonstrated to work [91] and has been further adapted and improved [25] in this way that the test is applicable on different vectors by means of appropriately determined index registers. It is not possible to read out all values at the end, but it is proposed as a possible replacement of QRAM in some cases [91]. Whether this parallel application can replace QRAM in the VQ application is an open question.

### 4.2. Winner Determination

Winner determination in prototype-based unsupervised and supervised vector quantization is one of the key ingredients for vector-shift-based adaptation for learning as well as median variants, which both inherently follow the winner-takes-all (WTA) principle (Equation 1). Obviously, the winner determination is not independent of the dissimilarity determination and, in quantum computing, is realized as a minimum search according to the list of all available dissimilarity values for a current system state.

An algorithm to find a minimum is the algorithm provided by Dürr and Høyer [92,93], which is, in fact, an extension of the often referenced Grover search [94]. Another sophisticated variant for minimum search based on a modified swap test, a so-called *quantum phase estimation* and the Grover search has been proposed [95]. Connections to the similar *k*-nearest neighbor approach were shown [96].

### 4.3. Updates Using Vector Shift

The normalization of quantum states places them on a hypersphere; this allows the transfer of the spherical linear interpolation (SLERP) to a quantum Computer [25]. This method is known as qSLERP, and the respective circuit is depicted in Figure 4. The qSLERP-circuit takes the two vectors |x〉 and |w〉 as input as well as the angle θ between them, which can be derived from the inner product and the interpolation position. The ancilla bit is measured, and the result in the data register is only kept if the ancilla is in the zero state. To store the result, the probability of the state of the data register has to be determined using repeated execution of the circuit.

From a mathematical point of view, the qSLERP approach is similar to the update used in Angle-LVQ [59] for non-quantum systems.

### 4.4. Median Adaptation

A selection task based on distances in median approaches is the Max–Sum Diversification problem; it can be mathematically transformed into an equivalent Ising model [97]. Other median approaches in VQ depend on the EM algorithm, like median *k*-means (*k*-medoids). A quantum counterpart of expectation maximization [98] was introduced as an extension of the *q*-means [99], a quantum variant of *k*-means. The authors showed the application of a fitting Gaussian Mixture Model. A possible generalization to other methods based on EM needs to be verified.

### 4.5. Vector Quantization as Set-Cover Problem

Above, in Section 2.1.3, we introduced the set-cover problem for unsupervised vector quantization. The QUBO model is NP-hard. Hence, at least in theory, the NP-complete set-cover problem can be transformed into it. A transformation from a (paired) set cover to the Ising model and, therefore, to QUBO can be solved with AQC [100]. Taking the view of vector quantization, the following transformation of an unsupervised ϵ-ball set-cover problem to a corresponding QUBO formulation can be done [77]:

Let $\left\{B_{\epsilon }\left(x{}_{i}\right)\right\}$ with i∈{1,⋯,N} be the set of ϵ-balls surrounding each data point $x{}_{i}\in X$. We introduce binary indicator variables $z_{i}$, which are zero if $B_{\epsilon }\left(x{}_{i}\right)$ does not belong to the current covering, and it is one elsewhere. Further, let $c_{k}$ be the number of sets $B_{\epsilon }\left(x{}_{i}\right)$ with $z_{i}=1$ and $x{}_{k}\in B_{\epsilon }\left(x{}_{i}\right)$, i.e., $c_{k}$ counts the number of covering ϵ-balls in the current covering. In the next step, we code the integer variables $c_{k}$ using binary coding according to let $c_{k,m}=1$ iff $c_{k}=m$ and zero otherwise. We impose the following constraint
$$\sum_{m=1}^{N}c_{k,m}=1:\forall k\;,$$
reflecting that the binary counting variables are consistent, and exactly one is selected. The second constraint establishes logical connections between the selected sets in the considered current covering and the counting variables by requiring that
$$\sum_{i|x{}_{k}\in B_{\epsilon }\left(x{}_{i}\right)}z_{i}=\sum_{m=1}^{N}m\cdot c_{k,m}\;:\;\forall k\;,$$
where m≥1 ensures that every point is covered. These constraints can be transformed into penalty terms using the squared differences between the left and the right side for each. Then the clustering task is to minimize the sum of all indicator variables $z_{i}$, taking the penalty terms into account. Using the explained construction scheme, this resulting cost function only contains pairwise interactions between binary variables without explicit constraints. Therefore, the set-cover problem is transformed into a QUBO problem.

Analog considerations are valid for the supervised classification task.

### 4.6. Vector Quantization by Means of Associative Memory

One of the first quantum associative memories based on a Hopfield network (HN) approach was proposed in 2000 [69]. Recently, a physical realization based on an actual quantum processor was provided [101]. As shown before, the HN energy function is identical to the QUBO problem, which can be solved by applying the quantum strategies in Section 4.7. Further, AQC for VQ was proposed, using HNs as an intermediate model [49].

A connection between gate-based quantum computing and HNs can be shown [102]. There, a solver based on Hebbian learning and mixed quantum states is introduced. The connection to complex-valued HN, as discussed in Section 2.1, is straightforward.

### 4.7. Solving QUBO with Quantum Devices

While we transformed most problems into QUBO in the previous subsections, we now connect them to quantum computing. Different strategies based on quantum computing hardware are available to solve QUBO problems. Heuristic approaches exist for many commercially available hardware types, from quantum annealers and gate-based computers to quantum devices based on photons.

Solve QUBO with AQC

A commercial approach in quantum annealing to solve QUBO or Ising models is described in the white paper [103] using the Company D-Wave. The solving of QUBO problems is the major optimization problem that is proposed to run on the limited hardware of a quantum annealer. According to this, the binary variables are physically implemented as quantum states. Values of the model interactions are implemented using couplers between pairs of qubits. Restrictions of the hardware make it necessary to order and map the qubits accordingly. The major open question about AQC is whether the length of the period grows slowly enough to be feasible.

Solve QUBO with Gate-Based Computing

For gate-based quantum computers, a heuristic called QAOA can approximately solve QUBO problems [104]. It contains two steps, first, optimizing a variational quantum circuit and second, sampling from this circuit. The ansatz of this circuit is a parametrized alternating application of the problem Hamiltonian and a mixing Hamiltonian. The expected value of the state gets then minimized using a classical computer, and different strategies have been proposed. With the found (local) minima, the quantum circuit gets executed, and the output gets sampled. Heuristically, low-energy states have a high chance of being sampled. It should be emphasized that it remains to be proven that QAOA has a computational advantage for any type of problem.

Solve QUBO with Photonic Devices

Gaussian Boson Sampling is a tool realized using quantum photonic computers, a kind of quantum hardware that has potential physical benefits that could lead to fast adoption. Quantum photonic devices introduce new types of quantum states into the field of quantum computing, like Fock states or photon counts. Gaussian Boson Sampling is seen as a near-term approach to utilizing quantum photonic computers. A solving strategy for QUBO by means of an Ising model taking a hybrid approach using Boson-sampling has been presented [105].

### 4.8. Further Aspects—Practical Limitations

Impact of Coding

We can replace all steps in the vector shift variant of VQ with quantum routines, but it is not possible to build up a complete algorithm so far. The main difficulty is that these atomic parts do not share the same encoding.

One example of this fact is the SWAP-test: Here, the result is stored as the probability of a qubit being in state |0〉. However, we have to get rid of the phase information to obtain a consistent result. Otherwise, this could lead to unwanted interference. A possible solution could be the exploration of routines based on mixed quantum states. However, the use of a Grover search is inconvenient for this task because it is based on basis encoded values, while the dissimilarity measures are stored as probabilities.

Impact of Theoretical Approximation Boundaries and Constraints

Some algorithms use probability or state estimation with sampling because it is impossible to directly observe a quantum state. For example, the output of the SWAP test has to be estimated using repeated measurements. The problem with an estimation of a measurement probe is well-known [25,90]. The field of finding the best measurement strategy for state estimation is called quantum tomography.

Another theoretical boundary is the loading of classical data to a real quantum device. Initializing an arbitrary state efficiently would be possible within the framework and regarding the implementation of the QRAM concept. However, the efficiency of those approaches is demanded because of the repeating nature of most algorithms and from the perspective of the non-cloning theorem.

Impact of Noisy Circuit Execution

The noisy nature of the current quantum hardware defeats most, if not all, of the theoretical benefits of quantum algorithms. A combination of improved hardware and quantum error correction will potentially solve this issue, allowing large-scale quantum computers.

## 5. Conclusions

The abstract motif of vector quantization learning has several adaptation realizations based on distinct underlying mathematical optimization problems. Vector shifts in prototype-based vector quantizers frequently are obtained as gradients of respective cost functions, whereas set-cover problem-related optimization belongs to binary optimization. Associative memory recalls rely on attractor dynamics. For these diverse paradigms, we highlighted (partially) matching quantum routines and algorithms. Most of them are, unfortunately, only heuristics. Further, their advantages over classical approaches have not been proven in general. However, the wide range of quantum paradigms, quantum algorithms, and quantum devices capable of assisting vector quantization translates into a broad potential of vector quantization for quantum machine learning. It is not possible to predict which quantum paradigm will succeed in the long term. Therefore, there is no outstanding vector quantization approach for quantum computing at the moment. But because many of the presented approaches can be transformed into QUBO problems, improved quantum solvers of each paradigm would have a strong impact. Especially, discrete strategies like median vector quantization, which are heavily restricted by classical computers, could become feasible. In other words, if a quantum advantage can be demonstrated in the future, vector quantization will likely benefit, but the direction will be set with improvements in the construction of quantum devices.

Finally, we want to emphasize that the overview in the paper is not exhaustive. For example, a possible connection that was not introduced above is the use of the probabilistic nature of quantum computing in combination with the probabilistic variants of Learning Vector Quantization [106].

However, we also should mention that the question of possible quantum supremacy, or even quantum advantages, is currently still considered an open problem in the literature. It has been discussed to be merely a weak goal for quantum machine learning [107]. Due to the lack of the existence of sufficient hardware today, it is also not possible to compare real runtimes adequately.

Nevertheless, the theoretical understanding of the respective mathematical concepts and their physical realization is important for progress in quantum computing and, hence, also in quantum-related vector quantization. 

## Figures and Tables

**Figure 1 entropy-25-00540-f001:**
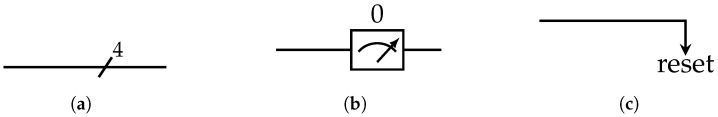
Example for notations. (**a**) Quantum Register. (**b**) Branching Measurement. (**c**) Garbage State.

**Figure 2 entropy-25-00540-f002:**
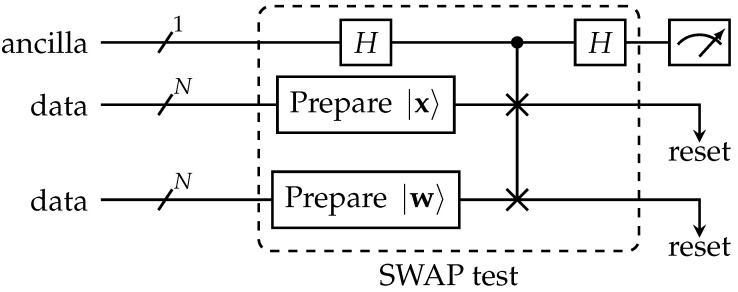
The SWAP-test to measure the dissimilarity between the states |x〉 and |w〉 using the ancilla qubit.

**Figure 3 entropy-25-00540-f003:**
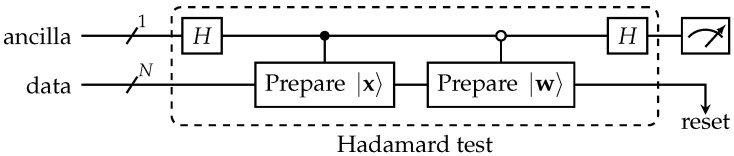
The Hadamard test to measure the dissimilarity between the states |x〉 and |w〉 using the ancilla qubit.

**Figure 4 entropy-25-00540-f004:**
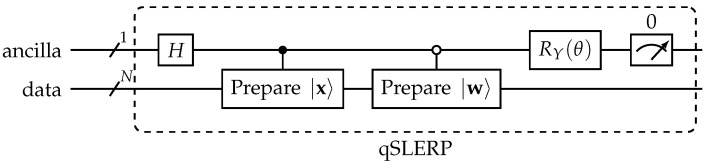
qSLERP circuit.

## Data Availability

Not applicable.

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
