# Peer review of "Quantum Computing Approaches for Vector Quantization—Current Perspectives and Developments"

_entropy, 2023, doi:10.3390/e25030540_

Round 1
Reviewer 1 Report
1. Only 4/20 pages (Section 4) describe the proposed scheme, and the rest are almost entirely about existing knowledge. This seriously affects the novelty and quality of the manuscript.
2. The lack of detail in the description in Section 4 prevents me from having a clear picture of the author's scheme. I cannot evaluate the correctness of it.
Author Response
1. only 4 of 20 papges describe the proposed scheme ...
The reviewer is right that this main chapter has only four papges.
However, the chapters 2 and 3 are important to motivate the chapter
4 and to guide the potential reader why these approaches are worth
to be considered.
2. lack of details
the aim is not to explain all technical details. Instead, we want to
point out which technical realization in quantum computing already
exist to tackle the problems/paradigms in standard vector quantization by quantum computing alternatives.
Reviewer 2 Report
This article provides an overview of vector quantization methods in classical machine learning as well as in quantum computing. Some suggestions and questions are given below.
(1) Please keep the reference format uniform.
(2) The abstract points out the importance of the VQ approach, but lacks use cases in the introduction section to demonstrate the importance of the VQ scheme.
(3) The abstract states that VQ methods are interpretable, while the quantum scheme projects data into Hilbert space through quantum circuits, which is called quantum feature mapping and sometimes understood as quantum kernel methods, is the quantum scheme still interpretable?
(4) There are various quantum schemes for VQ, but the lack of comparison does not allow us to know which of the so many approaches has more potential and value for research.
Author Response
- keep the references in an uniform format ...
we did it ... - Lack of use cases to underly importance of vecor quantization
We now mention several vector quantization models developed for
specic task raning from data compression and clustering to classi-
cation learning and task dependent feature relevance learning, which
are explicitly based on vector quantization schemes see introduction - Interpretability regarding quantum feature mapping
We agree with the reviewer that this is a crucial point. Now we mention that interpretability may be reduced or limited by the feature
mapping regarding the original data space. The interpretability in
the feature space, however, remains valid. see section 4.1 - Comparison of the approaches
Thanks for this hint: Generally, it is task dependnent which scheme/approach should be favored. All paradigms have their own merits and disadvantages. Thus a general statement regarding any 'best choice' is
in our opinion impossible. Yet, we included a small comparison,
which we relate to the adaptation schemes vor vector quantization
approaches in the introduction.
Reviewer 3 Report
A nice overview paper. Some remarks.Page 10, line 367. I would indicate that since garbage states are entangled they cannot be reused. Usually one resets them by un-computation. (Usually setting them to zero)
I would make a stronger division between the quantum gate model and adiabatic quantum computation. A quantum annealer such as D-Wave is only used for optimization task and is based on adiabatic quantum computation but can only be used for optimization tasks and has not the same power as a quantum circuit model.
Author Response
1. regarding entangled garbage states ...
Thanks for this comment. We included your phrases into the text.
2. Stronger division between quantum gate model and quantum annealer
Thanks for this suggestion: we included a respective comment at the
end of sect. 3.2.1.
Round 2
Reviewer 1 Report
The two issues I mentioned last time have not been resolved.
1. Only 4/20 pages (Section 4) describe the proposed scheme, and the rest are almost entirely about existing knowledge. This seriously affects the novelty and quality of the manuscript.
2. The lack of detail in the description in Section 4 prevents me from having a clear picture of the author's scheme. I cannot evaluate the correctness of it.
Author Response
We thank the reviewers for mostly accepting the revisions we made. For the
remaining problems according to the opinion of the rst reviewer, we repeat
our arguments:
1. only 4 of 20 pages describe the proposed scheme ...
The reviewer is right that this main chapter has only four pages.
However, the chapters 2 and 3 are at least as important as the chapter 4 because these chapters to guide the potential reader why the
considered approaches are worth to be reviewed.
2. lack of details
again, the aim is not to explain all technical details of the mentioned approaches. Rather, we want to point out which technical
realization in quantum computing already exist to tackle the problems/paradigms and, particularly, the standard update schemes in
vector quantization by quantum computing alternatives.
We hope to convince the reviewers with these arguments.
Reviewer 3 Report
The paper is ready for publication
Author Response
Thank you again for your responses.